# A solvent-free processed low-temperature tolerant adhesive

Xiaoming Xie[1,2], Yulian Jiang[2], Xiaoman Yao[1], Jiaqi Zhang[3], Zilin Zhang[2], Taoping Huang[1], Runhan Li [1] ✉, Yifa Chen [1] ✉, Shun-Li Li[1] & Ya-Qian Lan [1] ✉

Ultra-low temperature resistant adhesive is highly desired yet scarce for material adhesion for the potential usage in Arctic/Antarctic or outer space exploration. Here we develop a solvent-free processed low-temperature tolerant adhesive with excellent adhesion strength and organic solvent stability, wide tolerable temperature range (i.e. −196 to 55 °C), long-lasting adhesion effect ( > 60 days, −196 °C) that exceeds the classic commercial hot melt adhesives. Furthermore, combine experimental results with theoretical calculations, the strong interaction energy between polyoxometalate and polymer is the main factor for the low-temperature tolerant adhesive, possessing enhanced cohesion strength, suppressed polymer crystallization and volumetric contraction. Notably, manufacturing at scale can be easily achieved by the facile scale-up solvent-free processing, showing much potential towards practical application in Arctic/Antarctic or planetary exploration.

Adhesive plays a critical role in numerous fields such as construction, textile, electronic products and aerospace, etc[1-3]. The ever-growing practical demands and sustainable development for society and industry within a wide temperature range, for example, research at the poles of the Earth (e.g., South Pole, 20.7 to −94.2 °C) and human space exploration (e.g., Moon, 127 to −183 °C; Mars, 20 to −140 °C; Saturn, −130 to −191 °C; Neptune, −210 to −218 °C), call for high-strength adhesive at low temperature[4-8]. Up to now, a majority of traditional adhesives are based on polymer as the main component[9-11], for example, commercially available hot melt adhesives include ethylene-vinyl acetate copolymer (EVA), polyamide (PA) and polyether sulfone (PES), etc. Despite their widespread use in daily life, they still have some bottlenecks[12-17], especially at low temperature: (1) high cross-linking density and low surface energy lead to difficult bonding and easy debonding between substrate surface and adhesive; (2) poor interfacial infiltration effect with easily formed thicker adhering layer, resulting in undesired residual stress; (3) the traditional polymer molecules tend to be frozen at low temperature, leading to volumetric contraction, enhanced fragileness, weakened mechanical force transmission across the substrate, and reduced resistance to crack propagation and (4) the long-term stability in low temperature is generally

unmet, and the adhesion mechanism especially that under low temperature has been less investigated. Although some strategies, such as adding plasticizer/crosslinking agents or non-polar substituents, can elevate the temperature tolerance range of adhesives to some extent[15,17,18], the lowest temperature resistance for most of commercial hot melt adhesives is above −50 °C. Therefore, the novel functional polymer-based adhesives that can be used at ultra-low temperature are still demanded yet largely unmet for specific scenarios like Arctic/Antarctic or outer space exploration.

Polyoxometalates (POMs) have triggered a flurry of attention to the fields of chemistry and materials science owing to their unique physic-chemical properties[19-22]. Actually, POMs are frequently deemed as desired building blocks for constructing adhesive due to the following advantages[23-26]: (1) favorable interface adhesion might be promoted by regulating the crosslink density of polymer using POMs to reduce the residual stress; (2) cohesion strength of adhesive would be markedly enhanced by the interaction with POMs carrying multiple protons and oxygen-rich surface, resulting in the strong interaction energy and large energy dissipation[27-29] and (3) POMs with well-defined structures and compositions can act ideal templates for the theoretical calculations[30,31]. Nevertheless, the currently reported POMs-based

[1]School of Chemistry, South China Normal University, Guangzhou 510006, PR China. [2]Department of Chemistry, Xinzhou Normal University, Xinzhou, Shanxi 034000, China. [3]College of Physics and Optoelectronics, Taiyuan University of Technology, Taiyuan 030024, China. ✉e-mail: lirh949@m.scnu.edu.cn; chyf927821@163.com; yqlan@m.scnu.edu.cn

adhesives usually rely on solvents (e.g., organic or aqueous agents)[32–37], since it can break intermolecular forces of POMs and other counterparts, and subsequently assemble into viscous materials[38,39]. However, low temperature would lead to phase transition of solvents and media, which makes adhesive become brittle or deformed to seriously weaken their functionality or applicability[5]. Up to now, it has been reported that POMs-based adhesive can act as low-temperature adhesion by removing the solvent[40], yet the presence of solvents would result in many inconveniences in practical applications including storage, transportation, or processing processes, as well as the possible performance suppression caused by the residue of solvents. Thus, based on the above considerations and inspired by the pioneering works, the exploration of solvent-free method to facilely prepare low-temperature tolerant adhesive would be an intriguing target for practical usage like Arctic/Antarctic or outer space exploration, yet related research works have been rarely reported as far we know.

As a proof-of-concept, a type of $H_4SiW_{12}O_{40}$ ($SiW_{12}$) based solvent-free polymer (SSFP) adhesive has been successfully designed and prepared (Fig. 1a, b). The SSFP adhesive exhibits high adhesion strength, favorable interfacial adhesion ability, excellent organic solvent stability, and ultra-low-temperature tolerance, which is superior to commercially available hot melt adhesives. Moreover, theoretical calculations prove that the strong interaction energy between $SiW_{12}$ and polyethylene glycol (PEG) through abundant hydrogen bonds endows SSFP adhesive with favorable adhesion performance under a wide temperature range. This work may promote the development of solvent-free adhesives for potential applications of Arctic/Antarctic or planetary exploration.

## Results and discussion
### Synthesis and characterizations of SSFP adhesive

A white solvent-free SSFP adhesive is facilely prepared on a kilogram scale through a heat-assisted process (Supplementary Fig. 1, detail see "Methods" section). Fourier-transform infrared (FT-IR) spectra (Fig. 2a and Supplementary Fig. 2) show that four typical characteristic peaks of $SiW_{12}$ deriving from the stretching vibration bands of $W = O_d$, $Si-O_a$, $W-O_b-W$, and $W-O_c-W$, respectively, are still clearly discernible in the SSFP adhesive, indicating the retained structure of $SiW_{12}$ within the SSFP adhesive matrix. Simultaneously, the stretching vibration of etheric oxygen groups (−C−O−C−) in PEG at 1113 $cm^{-1}$ is slightly shifted to 1108 $cm^{-1}$ after forming SSFP adhesive[30], which is ascribed to the possible hydrogen-bonding interaction. Besides, the powder X-ray diffraction (PXRD) certifies the amorphous nature of SSFP adhesive (Fig. 2b), which is different from PEG and $SiW_{12}$, or even their physical mixture (Supplementary Fig. 3). These results suggest that $SiW_{12}$ is dispersed in SSFP adhesive matrix, and the crystallization of PEG is obviously inhibited after hybridizing with $SiW_{12}$. Additionally, their structures, chemical compositions and states of PEG and $SiW_{12}$ in adhesive have been confirmed with X-ray photoelectron spectroscopy (XPS) (Supplementary Figs. 4 and 5), $^{32}Si$ NMR (Fig. 2c) spectra[41], $^1H$ NMR spectra (Fig. 2d), solid-state and liquid-state $^{13}C$ NMR tests (Fig. 2e, f and Supplementary Fig. 6). The low field nuclear magnetic resonance (LF-$^1H$ NMR) reveals that a ~45 times decrease of crosslink density (from $69.93 \times 10^{-4}$ to $1.55 \times 10^{-4}$ mol mL$^{-1}$) can be detected in SSFP adhesive when compared with PEG (Fig. 2h), indicating that $SiW_{12}$ would be hybridized with PEG and occupies a certain space to decrease the number of cross-linked bonds in PEG. Moreover, the results are further supported by the decaying proton transverse relaxation curves (Fig. 2g)[42]. Besides, the scanning electron microscopy (SEM) tests show that SSFP adhesive has a denser and flatter surface than that of PEG treated under similar heating process (Fig. 3a and Supplementary Fig. 7), proving the vital role of $SiW_{12}$ in decreasing the residual stress. Furthermore, energy-dispersive X-ray spectroscopy (EDS) element mapping analyses indicate the uniform dispersion of $SiW_{12}$ in SSFP adhesive (Fig. 3a).

The adhesion effect plays a vital role in the application of adhesive materials[43]. Satisfyingly, this obtained SSFP adhesive (Fig. 3b) exerts favorable adhesion capability on different types of artificial and natural

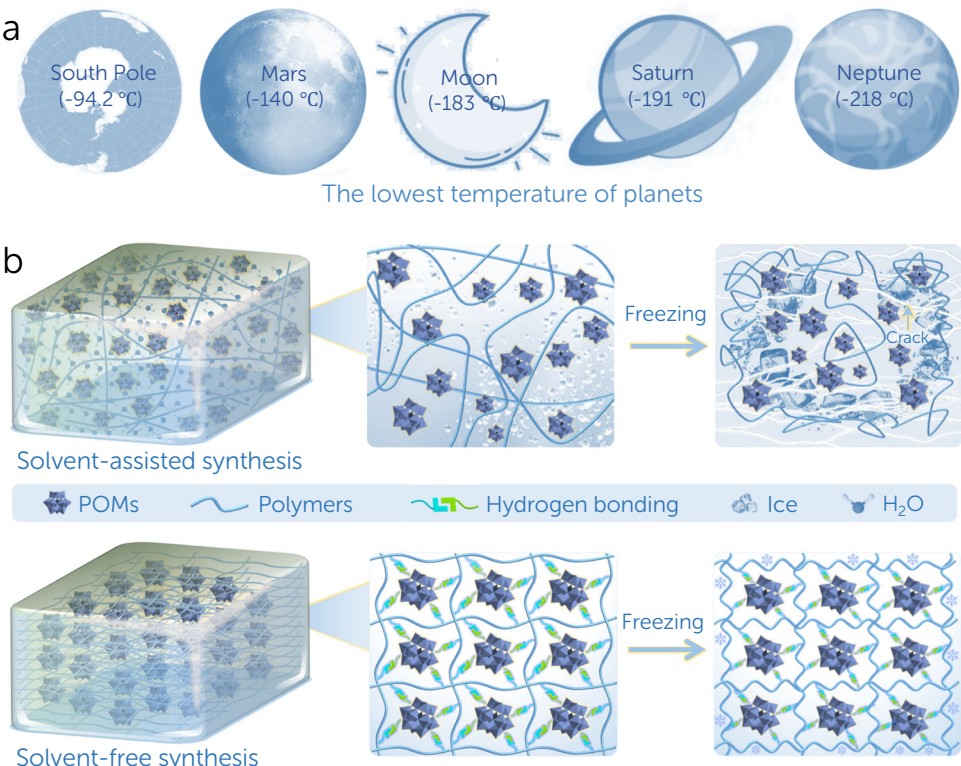

Fig. 1 | **Schematic illustration of the low-temperature effect on adhesion behaviors. a** The lowest temperatures of the South Pole and representative outer space planets. **b** The schematic illustration of the low-temperature effect on adhesion behaviors for the solvent-assisted and solvent-free POMs-based adhesives.

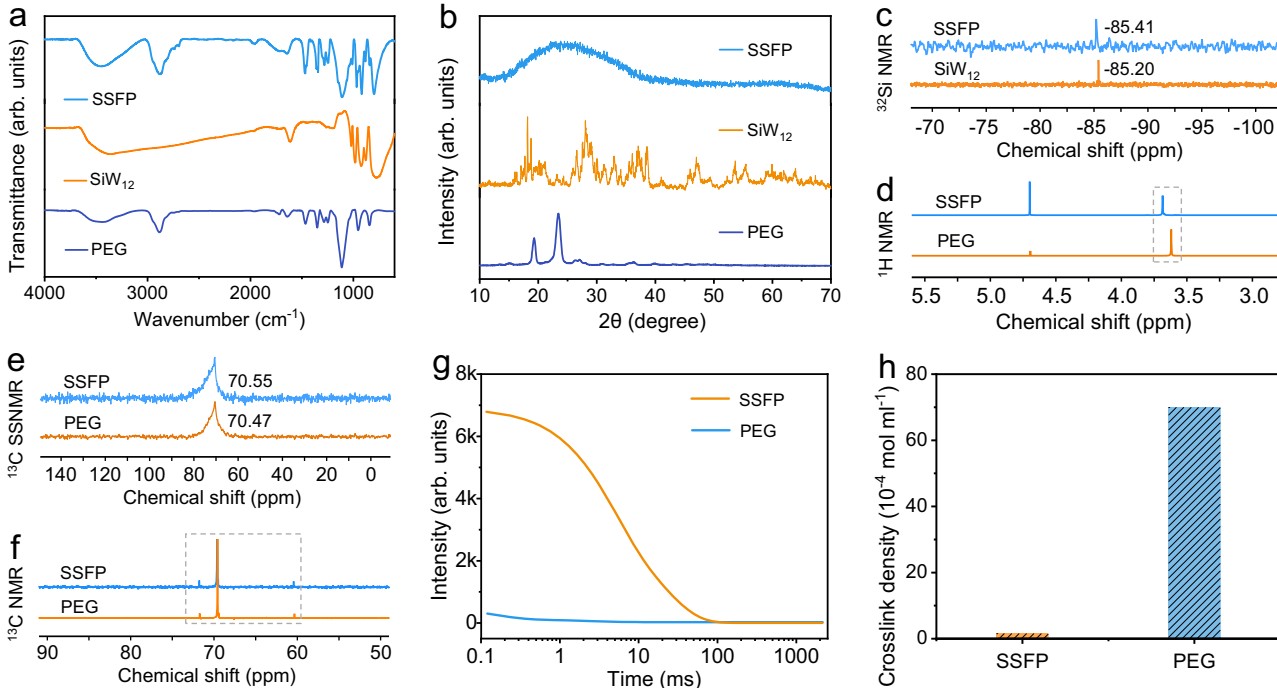

**Fig. 2 | Characterization of the SSFP adhesive. a** FT-IR spectra of SSFP, SiW$_{12}$, and PEG. **b** PXRD patterns of SSFP, SiW$_{12}$, and PEG. **c** $^{32}$Si NMR spectra of SSFP and SiW$_{12}$. **d** $^1$H NMR spectra of SSFP and PEG. **e** $^{13}$C CP-MAS NMR spectra of SSFP and PEG. **f** $^{13}$C NMR spectra of SSFP and PEG. **g** Proton transverse relaxation curves of SSFP and PEG. **h** Crosslink densities of SSFP and PEG.

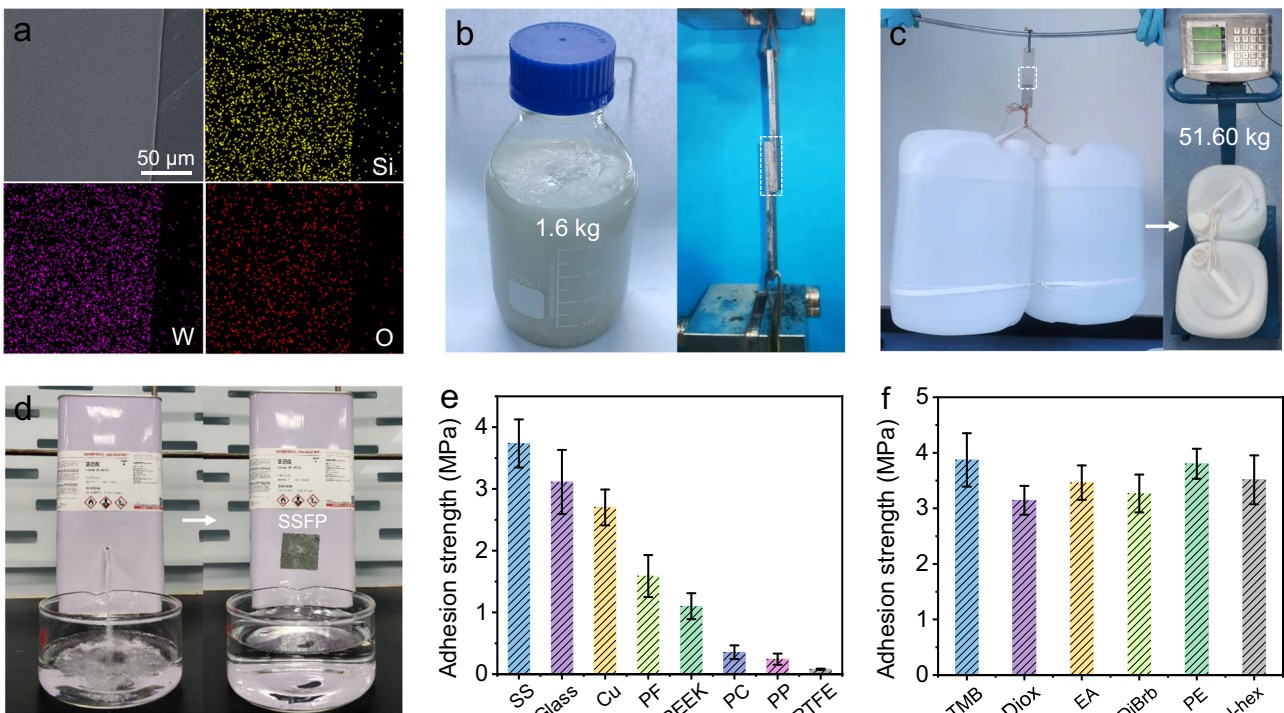

**Fig. 3 | Morphology and strength property of SSFP adhesive. a** SEM and corresponding elemental mapping images of SSFP adhesive. **b** Macroscopic photographs of SSFP adhesive obtained in kilogram scale and shear strength test. **c** Photograph of the weights bonded by SSFP adhesive. **d** Emergency leakage test performed using SSFP adhesive. **e** Adhesion strengths of SSFP adhesive on various substrates. **f** Adhesion strengths of SSFP adhesive on the interfacial adhesion system in various organic solvents for 14 days. The error bars for (**e**, **f**) represent mean ± standard deviation (*n* = 3 independent samples).

materials (Supplementary Fig. 8). Subsequently, the quantitative tests of adhesion strength of SSFP adhesive (Fig. 3e) are measured. Strong adhesion forces are presented on high surface energy substrates, such as stainless steel (SS, 3.7 MPa), glass (3.1 MPa), and copper (Cu, 2.7 MPa), owing to the existence of strong chemical bonds and mechanical interlocking[44]. Notably, the SSFP adhesive with favorable adhesion performance is superior to almost all of the POMs-based adhesives and comparable to most of solvent-free adhesives reported so far (Supplementary Fig. 9, Supplementary Tables 1 and 2)[30,33–36,38,41]. Meanwhile, relatively weaker adhesion forces are shown on low surface energy substrates, such as polycarbonate (PC), polypropylene (PP), and polytetrafluoroethylene (PTFE) (Fig. 3e). Besides, the SS substrates joined with SSFP adhesive can easily tolerate a weight of ~50 kg (Fig. 3c), indicating the remarkable adhesion capabilities of SSFP adhesive. Furthermore, it can be observed that the adhesion strength of SSFP adhesive on SS still maintains at about 3.7 MPa after multi-recyclable adhesion and deadhesion recycling (Supplementary Fig. 10). The distribution of SSFP adhesive on SS after detachment implies that the adhesion failure mainly occurs in interfacial adhesion between adhesive and substrate, indicating the high cohesion interaction of SSFP adhesive (Supplementary Fig. 11)[45,46].

Based on the above results, SS substrate is selected as a model substrate to further monitor the adhesion performance and elucidate the interaction mechanism of SSFP adhesive. Interestingly, the adhesion strength of SSFP adhesive on SS is positively proportional to $SiW_{12}$ content (mass ratio of $PEG/SiW_{12}$, 2:5–2:3) (Supplementary Figs. 12–16). Nevertheless, the adhesive will transform into a very hard and brittle material with negligible viscosity when adding excessive $SiW_{12}$. To elucidate the effect of the molecular weight of PEG on its adhesive property, the molecular weight of PEG has been screened from ~2000 to ~20000 based on its rigidity (strength) and viscosity in macroscopic level. Results demonstrate that all of these PEG polymers can form adhesives with $SiW_{12}$ (Supplementary Fig. 17), and the adhesion strengths gradually enhance with the increasing molecular weight of PEG (Supplementary Fig. 18). The optimal viscosity in the macroscopic level can be achieved for PEG (~10000), which we believe there might exist a balance between the viscosity and hardness of adhesive, and polymers with higher molecular weight are not conducive to the improvement of viscosity. Therefore, the SSFP adhesive with the optimized composition (PEG, ~10000; mass ratio = 2:5) is selected as a model sample for subsequent in-depth study. It is noteworthy that the adhesion strength of SSFP adhesive is far superior to the $SiW_{12}$-based solvent-assisted polymer (SSAP) adhesive[30] (approximately 65 times, Supplementary Figs. 19 and 20). As expected, a similar $H_3PW_{12}O_{40}$ ($PW_{12}$) based adhesive (PSFP) can also be produced by the same solvent-free method (Supplementary Fig. 1) and has been confirmed by various characterizations (Supplementary Figs. 21–23). Whereas, the adhesion strength of PSFP adhesive is weaker than that of SSFP adhesive, which testifies that $SiW_{12}$ with more protons is more conducive to the formation of high-strength adhesives (Supplementary Fig. 16). Additionally, no adhesive is observed by heat-assisted process after replacing $PW_{12}$ with $Na_3PW_{12}O_{40}$ (Supplementary Figs. 24–26), confirming the vital role of protons for the preparation of solvent-free adhesives. In addition, the influence of polymer type on the formation of adhesive has been investigated by replacing PEG with polytetramethylene glycol (PTMEG), polypropylene glycol (PPG), polycaprolactone (PCL), polyethylene (PE), and polyvinylidene difluoride (PVDF). The results display that adhesives can still be formed when replacing PEG with PTMEG, PPG, and PCL, yet the adhesion strengths are weaker than that of SSFP adhesive, which might be the less hydrogen bond acceptors in the alternative polymer chains or their own physical properties (Supplementary Figs. 27–29). Besides, no adhesive has been formed when replacing PEG with polyethylene (PE) and polyvinylidene difluoride (PVDF) (Supplementary Fig. 27), manifesting the irreplaceable important role of hydrogen bond acceptor for

the formation of solvent-free adhesives. Beyond that, the PEG analogs, PEGME bearing the methyl group and hydroxyl group at each terminus and PEGdME bearing the methyl groups at both termini, can also crosslink with $SiW_{12}$ to form adhesives (Supplementary Fig. 30), and the rheology and lap-shear adhesion test results show that these adhesives possess similar shear strength (Supplementary Fig. 31) and viscosity (Supplementary Fig. 32), suggesting that the formation and behavior of adhesive are primarily attributable to hydrogen bond interaction between protons of $SiW_{12}$ and etheric oxygen groups of PEG rather than the terminal groups.

As is well-known, most of adhesives based on polymers tend to be dissolved or swelled in organic solvents, resulting in markedly weakened adhesion performances and seriously hindered practical applications[47]. Hence, the maintenance of robust adhesion strength in organic solvents is an eye-catching trait for adhesives. Interestingly, the SSFP adhesive is insoluble in some organic solvents, such as mesitylene (TMB), dioxane (Diox), octanoic acid (OA), 1,4-dibromo-butane (DiBrb), petroleum ether (PE) and N-hexane (N-hex). For example, no separation or displacement phenomenon has been observed in a long-term adhesion test for at least 14 days in N-hex (Supplementary Fig. 33). In addition, the adhesion strengths of adhesive remain relatively stable for 14 days after immersion in these organic solvents (Fig. 3f) Moreover, SSFP adhesive is capable of performing rapid adhesion (Supplementary Fig. 34 and Supplementary Movie 1) and preventing an emergency leakage for organic solvents (Fig. 3d and Supplementary Movie 2). In sharp contrast, ethyl vinyl acetate (EVA), a kind of commercially available hot melt adhesive, shows negligible adhesion strength after soaking in mesitylene for only 7 days when compared to that of SSFP adhesive (Supplementary Fig. 35). Additionally, the resistance of SSFP adhesive to water has been investigated through both water immersion and humidity resistance experiments. By detecting the UV-vis absorption of $SiW_{12}$ in the soaked solution, we find that only minor amount (0.76%) SSFP adhesive is dissolved into the water after soaking for 5 days (Supplementary Figs. 36 and 37). However, it becomes soft and partly transforms into traditional solvent-assisted adhesive, which would be attributed to the interaction of water molecules with the adhesive that results in parts of the hydrogen-bonded networks in SSFP adhesive to be collapsed and rebuilt. Furthermore, the adhesion strengths of SSFP adhesive at different relative humidity (from 40% to 80%) have also been tested, and the results show that the adhesion strengths of SSFP adhesive only have slight decrease with the increase of relative humidity (Supplementary Fig. 38), proving its high resistance to humidity.

Based on the above-mentioned high adhesion strength of SSFP adhesive, its viscosity, energy storage, and loss modulus have been further traced using rheology tests. Compared to the PEG, the SSFP adhesive exhibits higher viscosity and lower liquidity in rheological characterization (Supplementary Fig. 39). Rheology measurements verify that the modulus and viscosity are both inversely proportional to the operating temperature (Fig. 4a, b). The SSFP adhesive remains gel-like or solid-like state in macroscopic behaviors at below ~50 °C. Specifically, the loss modulus of the SSFP adhesive (Fig. 4a) exceeds storage modulus ($G'' > G'$) at above ~50 °C, resulting in a viscosity-dominated viscoelasticity state that can accelerate the interfacial bonding. Moreover, the reversibility of storage modulus ($G'$), loss modulus ($G''$), and complex viscosity ($\eta^*$) can be realized in cycling tests under circulating temperatures, which might be attributed to the solvent-free phase and invertible hydrogen bond interaction that enable reversible temperature-induced rheological behaviors.

## Ultra-low-temperature resistant performance
Viscous materials with low-temperature resistance play an important role in exploration under extreme environments, especially in a wide temperature-variable range, such as research at the poles of the Earth

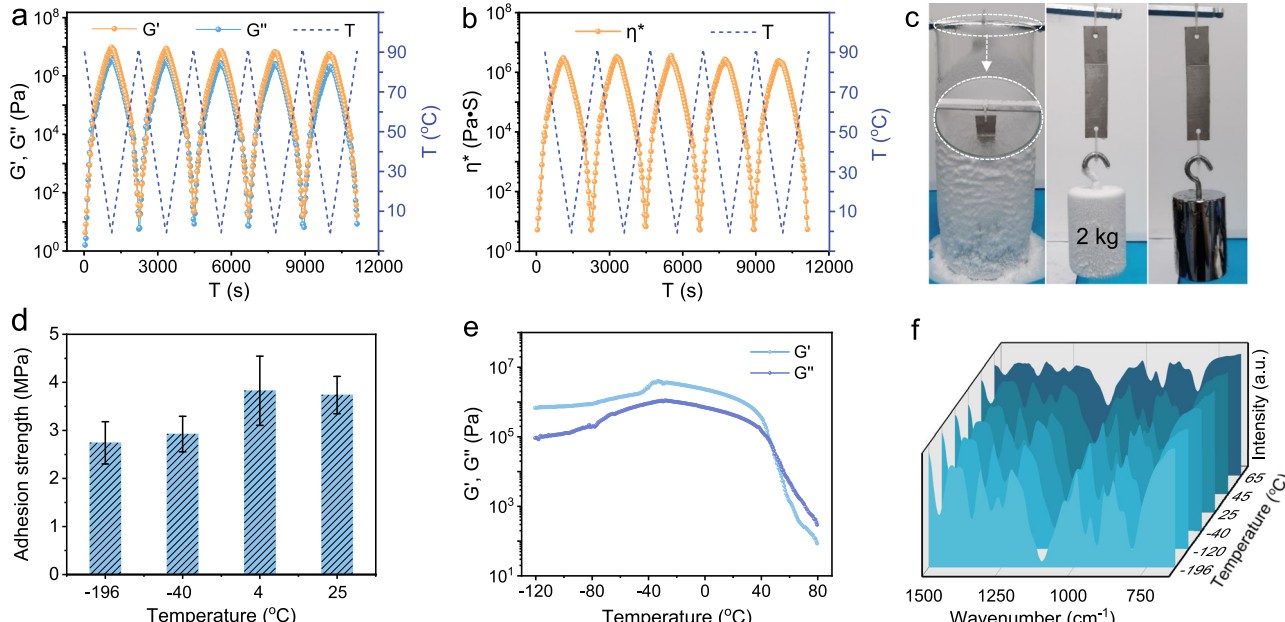

**Fig. 4 | The behaviors of solvent-free SSFP adhesive under a wide temperature range. a** The reversible curve of storage modulus (G′) and loss modulus (G″) of SSFP adhesive at cyclic temperature (T). **b** The reversible curve of complex viscosity (η*) of SSFP adhesive at cyclic temperature (T). **c** Macroscopic adhesion tests of SSFP adhesive in liquid nitrogen. **d** Adhesion strengths of SSFP adhesive on SS substrate at various temperatures. **e** Rheology measurements of SSFP adhesive between −120 and 80 °C. **f** Temperature-dependent FT-IR spectra of SSFP adhesive from −196 to 65 °C. The error bars for (**d**) represent mean ± standard deviation (*n* = 3 independent samples).

(e.g., the South Pole, 20.7 to −94.2 °C) and exploration of the outer planets (e.g., Mars, 20 to −140 °C). However, traditional polymer-based adhesives tend to become brittle and increase its residual stress as the temperature decreases[48]. Here, the thermogravimetric analysis (TGA) and differential scanning calorimeter (DSC) tests display that the glass transition temperature ($T_g$) of SSFP adhesive is −31.1 °C, which is lower than that of PEG (48.3 °C) (Supplementary Figs. 40 and 41), implying that the SSFP adhesive has better flexibility at relatively low temperature. Impressively, the SSFP adhesive adhered between SS slices can easily tolerate a 2 kg weight under liquid nitrogen conditions, and still maintain the original state after returning to room temperature (Fig. 4c and Supplementary Movie 3). In addition, the SSFP adhesive is not observed to have significant contraction or rupture after freezing with liquid nitrogen (Supplementary Fig. 42). In contrast, the obvious volumetric shrinkage and rupture for PEG and EVA adhesive occur at 25 °C and −196 °C, respectively (Supplementary Figs. 43 and 44). The above results show that the volumetric contraction of SSFP adhesive can be significantly inhibited at −196 °C after crosslinking SiW$_{12}$ with PEG. To support it, the adhesion strength of SSFP adhesive has been tested at different temperatures. Particularly, the adhesion strength is negligible for SSFP adhesive at 55 °C, then gradually increases as the temperature decreases to 4 °C (~3.8 MPa) (Fig. 4d and Supplementary Fig. 45). After that, the adhesion strength gradually decreases when the temperature decreases to −196 °C. More significantly, the SSFP adhesive can still maintain relatively high adhesion strength of 2.96 MPa after immersion in liquid nitrogen for 60 days (Supplementary Fig. 46), while the EVA adhesive is immediately frozen-cracked once it entered liquid nitrogen (Supplementary Fig. 47). These results demonstrate that the SSFP adhesive has admirable ultra-low-temperature resistant adhesion performance than that of commercial hot melt adhesive. Furthermore, the low-temperature rheological measurements (Fig. 4e) showed that SSFP adhesive has a stable storage modulus (G′) and loss modulus (G″) in the temperature range from −120 to 25 °C, which is consistent with its actual performance. Moreover, the temperature-dependent FT-IR spectra (Fig. 4f and Supplementary Fig. 48) indicate negligible change in the four signals of both SiW$_{12}$ and PEG,

manifesting that the interactions in SSFP adhesive remains almost intact under wide temperature range[49].

Based on the above excellent adhesion properties of SSFP adhesive, the adhesion mechanism has been further investigated. In general, cohesion and interfacial adhesion are two main factors affecting adhesion performance. Nevertheless, most of the theoretical calculation of adhesives focus on the simulation between adhesive and substrate, and it is still very scarce for the study of interaction contributing to cohesion. Thus, we have applied the density functional theory (DFT) calculations to investigate the SSFP adhesive at the molecular level[31,40]. The results reveal that the interaction energy between them is −287.7 kJ/mol for one PEG fragment, −537.4 kJ/mol for two PEG fragments, and −826.7 kJ/mol for three PEG fragments, respectively (Fig. 5a, Supplementary Figs. 49 and 50). Obviously, the binding stability between POMs and PEG can be significantly enhanced via strong hydrogen bond interaction. Hence, SiW$_{12}$ as a crosslinking agent will be interweaved and anchored into PEG networks, resulting in the formation of stable and durable adhesive. In addition, the molecular dynamics (MD) simulation is further performed to evaluate the temperature-dependent interaction energy and hydrogen bonds. At 25 °C, the interaction energy and hydrogen bond percentage between PEG and SiW$_{12}$ are average −1168 kJ/mol and 39.13% in this model system at 2 ns, respectively (Fig. 5b, Supplementary Fig. 51 and Supplementary Movie 4). At 55 °C, the fluctuation of interaction energy is more obvious and significantly reduces to −980 kJ/mol (Supplementary Figs. 52 and 53, Supplementary Movie 5). Moreover, the percentage of hydrogen bonds (40.0%) at 55 °C remains almost the same with that at 25 °C (Supplementary Fig. 54). Remarkably, at −196 °C, the interaction energy quickly reaches equilibrium and remain stable within 50 ps (Fig. 5c and Supplementary Movie 6). In addition, the dramatically increased interaction energy remains at average −1250 kJ/mol (Fig. 5d), meanwhile the percentage of formative hydrogen bonds (35.71%) is slightly affected by low temperature (Supplementary Fig. 51). High interaction energy at low temperature will elicit large energy dissipation of SSFP when dragging the adhesive, in which the synergistic interaction of them might be the dominating reasons for

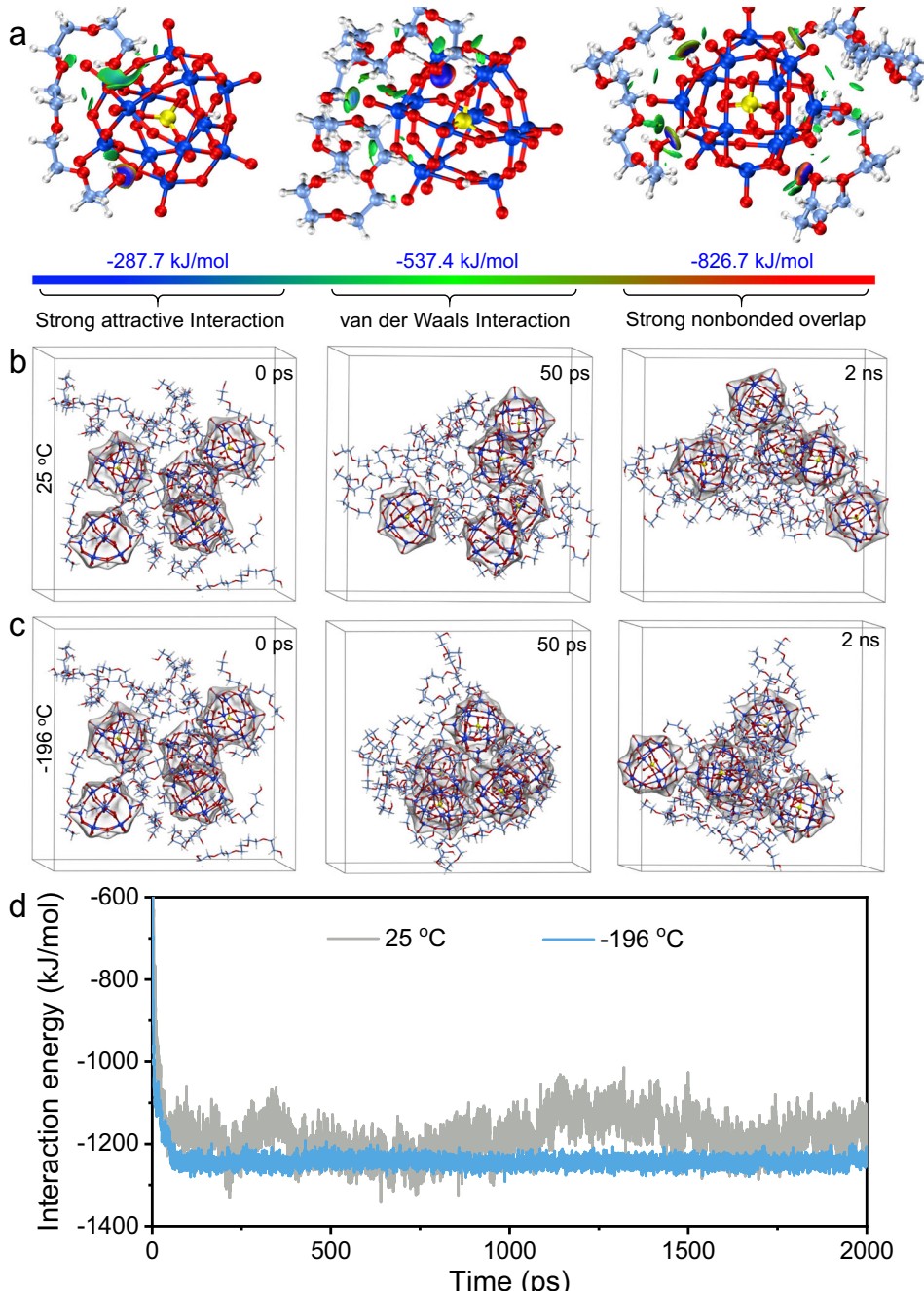

**Fig. 5 | Theoretical calculation for the interaction between SiW₁₂ and PEG.** **a** Schematic illustration of the interaction between SiW₁₂ and PEG for the adhesive formation (one, two, and three PEG fragments from left to right, respectively). **b** Snapshots of the aggregation behavior of SiW₁₂ and PEG at 25 °C. **c** Snapshots of the aggregation behavior of SiW₁₂ and PEG at −196 °C. **d** The interaction energy of PEG and SiW₁₂ during simulated process at 25 °C and −196 °C.

the SSFP adhesive to exhibit excellent adhesion performance at ultra-low temperature. As a contrast, when replacing SiW₁₂ with PW₁₂, both the interaction energy and the percentage of formative hydrogen bonds are significantly reduced to −817 kJ/mol and 29.27% (Supplementary Figs. 55−57), owing to the fewer protons that can be provided by PW₁₂ to crosslink with PEG under the same conditions. Furthermore, the MD simulations verify that there is favorable interfacial adhesion between SSFP adhesive and SS substrate, and the interaction energy increases gradually along with the decrease of temperature (Supplementary Fig. 58 and Supplementary Movies 7−9).

In summary, we have prepared a kind of POMs-based solvent-free polymer adhesive on a kilogram scale through a heat-assisted process.

It is worth noting that the achieved SSFP displays excellent interfacial adhesion ability on different substrates, high adhesion strength, and organic solvent stability, wide tolerable temperature range (i.e., −196−55 °C), and long-lasting adhesion effects (>60 days) at −196 °C. The high performance of SSFP exceeds that of commercial hot melt adhesives. Furthermore, combined experimental results with theoretical calculations, the strong interaction energy between POMs and PEG is the main factor for the high adhesion performance at low-temperature possessing enhanced cohesion strength, suppressed polymer crystallization, and volumetric contraction. This work enriches the types of low-temperature resistance adhesives and would shed light on the development of advanced solvent-free adhesives for Arctic/Antarctic or planetary exploration.

## Methods

### Syntheses of the SSFP adhesive with different mass ratio (2:3, 2:4, and 2:5)

The POMs-based solvent-free polymer adhesives are prepared using a simple method. PEG (200 mg) and SiW$_{12}$ (300, 400, and 500 mg) are mixed evenly in glass bottles, then the mixture is heated at 90 °C for 2 h. The SiW$_{12}$-based solvent-free polymer (SSFP) adhesive is obtained after cooling to room temperature. The SSFP adhesive can also be prepared on a kilogram scale through increasing raw materials in equal proportions. As a contrast, PW$_{12}$-based solvent-free polymer (PSFP) adhesives are obtained by replacing SiW$_{12}$ with PW$_{12}$ under the same preparation method. In addition, The POMs-based solvent-assisted polymer adhesives (SSAP and PSAP) are prepared by adding 4 mL of water into the mixture of POMs and PEG, then standing for 2 h. (PEG:POMs = 2:5).

## Data availability

The data that support the findings of this study are available within the paper and its supplementary information files or are available from the corresponding authors upon request. Source data are provided with this paper.

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

## Acknowledgements

This work was financially supported by the National Key R&D Program of China (2023YFA1507204, Y.-Q. L.), NSFC (Grants 22225109, Y.-Q.L.; 22071109, S.L.L.; 22171139, Y.F.C.), Natural Science Foundation of Guangdong Province (No. 2023B1515020076, Y.F.C.) and Fundamental Research Program of Shanxi Province (20210302124339, X.M.X.).

## Author contributions

Y.-Q.L., Y.C., and X.X. conceived and designed the idea. X.X., R.L., and Y.C. designed the experiments, collected and analyzed the data. X.X., Y.J., J.Z., X.Y., Z.Z., and T.H. papered the experiments and characterizations. R.L. analyzed DFT and MD calculations. X.X., S.L., Y.C., and Y.-Q.L. discussed the results and prepared the manuscript. X.X. wrote the manuscript.

## Competing interests

The authors declare no competing interests.
