## [Peer Review File · Nature Communications]

REVIEWER COMMENTS

Reviewer #1 (Remarks to the Author):

This manuscript reported an adhesive with ultra-low temperature resistant. The experiment procedure is facile and multifunctional properties are interesting. But some issues need to be addressed before the manuscript becomes suitable for publication. The comments are stated below

1. The polymers showed important influence on the properties of the adhesive. How about replacing PEG with other polymers? Such as PTMEG or PPG (polypropylene glycol)? Does the molecular weight of PEG have effect on its adhesive strength?
2. The obtained adhesive showed adhesive organic solvent stability? I am wondering does the adhesive dissolve or swell in good solvent for PEG, such as water, THF, DMF and so on.

Reviewer #2 (Remarks to the Author):

A low-temperature tolerant adhesive was prepared through a heat-assisted process in this work. Because of hydrogen bonds between $\text{H}_4\text{SiW}_{12}\text{O}_{40}$ and polyethylene glycol, solvent-free polymer (SSFP) adhesive showed an adhesion strength of 3.7 MPa towards stainless steel and 2.96 MPa at $-196\text{ }^\circ\text{C}$. However, the adhesion strength of SSFP adhesive was not high enough for engineering application, and the advantages of using polyoxometalates and solvent-free process have not been fully demonstrated. In summary, this work does not meet the requirements of innovation and importance to be recommended for publication in Nature Communication. Some comments are listed below.

- (1) SSFP adhesive showed an adhesion strength of 3.7 MPa towards stainless steel and 2.96 MPa at $-196\text{ }^\circ\text{C}$. However, the adhesion strength of SSFP adhesive was much lower than that of structural adhesive (generally higher than 15 MPa), which is not enough for application in Arctic/Antarctic or planetary exploration.
- (2) SSFP adhesive could not tolerate temperatures above $55\text{ }^\circ\text{C}$, which seriously affected the stability of SSFP adhesive during application.
- (3) Water is more common than organic solvents in daily life and in Arctic/Antarctic. The water resistance of SSFP adhesive should be investigated.

Reviewer #3 (Remarks to the Author):

Adhesives that can withstand cryogenic temperatures are in great demand for material bonding in

Arctic/Antarctic and outer space exploration, but their supply is scarce. In this study, the authors prepared kilogram-scale, solvent-free polymer adhesives based on POM by a heat-assisted process. The resulting SSFPs exhibit excellent interfacial adhesion capabilities on a variety of substrates and are notable for their high bond strength and organic solvent stability, wide allowable temperature range (i.e., -196 to 55°C), and long-term adhesion effectiveness (>60 days) at -196°C. The high performance of SSFPs is superior to commercial hot melt adhesives commercial hot melt adhesives. Furthermore, experimental results combined with theoretical calculations indicate that the strong interaction energy between POM and PEG is the main reason for the high adhesive performance at low temperatures with increased cohesive strength, reduced polymer crystallization, and volume shrinkage. This research sheds light on the development of more types of low temperature resistant adhesives and advanced solvent-free adhesives for Arctic and Antarctic and planetary exploration. The topics covered in this manuscript range from basic to applied, and are sure to be of interest to a diverse readership. Therefore, this manuscript should be accepted by Nature Communications. I would, however, like to express some concerns as follows. I hope the authors will address them appropriately before publication.

- 1) The phrase "H protons" appears several times, but I think just "protons" is fine.
- 2) The relationship between OPLS and UFF used in the MD simulation is unclear. Please explain in more detail.
- 3) Is it safe to use only NVT ensemble simulations? NVT would not be able to reproduce the change in density with temperature change. It would be better to do NPT ensemble as well.
- 4) I have understood the cohesive strength of the interaction between POM and PEG. However, experimentally, the authors have observed fracture at the adhesive interface, so it would be good to evaluate not only the cohesive energy but also the adhesive energy between the substrate and the adhesive in the simulation and compare it with the cohesive interaction energy.
- 5) It is clear from the MD simulation movies that the adsorption conformation of PEG around POM is really diverse; did the authors take that diversity into account in their DFT calculations?

Reviewer 1:

This manuscript reported an adhesive with ultra-low temperature resistant. The experiment procedure is facile and multifunctional properties are interesting. But some issues need to be addressed before the manuscript becomes suitable for publication. The comments are stated below.

1. The polymers showed important influence on the properties of the adhesive. How about replacing PEG with other polymers? Such as PTMEG or PPG (polypropylene glycol)? Does the molecular weight of PEG have effect on its adhesive strength?

Response: Thank you for your insightful comments. We have investigated the influence of other polymers on the formation of adhesive by replacing PEG with polytetramethylene glycol (PTMEG), polypropylene glycol (PPG), polycaprolactone (PCL), polyethylene (PE) and polyvinylidene difluoride (PVDF). The results display that adhesives can still be formed with PTMEG, PPG or PCL as alternative polymers, yet the adhesion strengths are weaker than that of SSFP adhesive, which might be due to the less hydrogen bond acceptors in the alternative polymer chains or their own physical properties (Supplementary Figs. S27 to S29). Besides, no adhesive has been formed when replacing PEG with polyethylene (PE) or polyvinylidene difluoride (PVDF) (Supplementary Fig. S27), manifesting the irreplaceable important role of hydrogen bond acceptor for the formation of solvent-free adhesives. The relative discussion has been added in the revised manuscript (page 6, line 12-21) and Supporting Information (page 33-35).

To elucidate the effect of molecular weight of PEG on its adhesive property, the molecular weight of PEG has been screened from ~2000 to ~20000 based on its rigidity (strength) and viscosity in macroscopic level. Results demonstrate that all of these PEG polymers can form adhesives with SiW₁₂ (Supplementary Fig. S17), and the adhesion strengths gradually enhance with the increasing molecular weight of PEG due to the increased physical entanglement (Supplementary Fig. S18). The optimal viscosity in macroscopic level can be achieved for PEG (~10000), which we believe there might exist a balance between the viscosity and hardness of adhesive, and polymers with higher molecular weight are not conducive to the improvement of viscosity. As a proof of concept, the adhesive composed of PEG (~20000) and SiW₁₂ becomes more brittle, which obviously affects its processibility and resistance to ultra-low temperature (Fig. R1). Therefore, the SSFP adhesive with the optimized composition (PEG, ~10000; mass ratio = 2 : 5) is selected as the model sample for further in-depth study.

Now we have added relative discussion in the revised manuscript (page 5, line 23-29; page 6, line 1-3) and Supporting Information (page 23-24).

Supplementary Figure 17. Digital photographs of the adhesives after heating at 90 °C for 2 h. a) PEG_{2k} and SiW₁₂. b) PEG_{4k} and SiW₁₂. c) PEG_{8k} and SiW₁₂. d) PEG_{10k} and SiW₁₂. e) PEG_{20k} and SiW₁₂.

Supplementary Figure 18. Adhesion strengths of the different molecular weights of PEG based adhesives on SS substrate.

Figure R1. Adhesion tests of the adhesive based on PEG_{20k} and SiW₁₂ in liquid nitrogen. a) Initial adhesion state. b-c) Frost-cracked state.

Supplementary Figure 27. Digital photographs of the samples after heating at 90 °C for 2 h. a) Adhesive based on PPG and SiW₁₂. b) Adhesive based on PTMEG and SiW₁₂. c) Adhesive based on PCL and SiW₁₂. d) Mixture of PE and SiW₁₂. e) Mixture of PVDF and SiW₁₂.

Supplementary Figure 28. Adhesion strengths of different polymer based adhesives on SS substrate. a) PPG. b) PTMEG. c) PEG (M_n, ~2000).

Supplementary Figure 29. Adhesion strengths of different polymer based adhesives on SS substrate. a) PCL. b) PEG (M_n, ~10000).

2. The obtained adhesive showed adhesive organic solvent stability? I am wondering does the adhesive dissolve or swell in good solvent for PEG, such as water, THF, DMF and so on.

Response: Thanks for your insightful comments. In this version, we have screened the chemical stability of adhesive in a series of organic solvents. The obtained adhesive shows high stability in some organic solvents, such as mesitylene (TMB), dioxane (Diox), octanoic acid (OA), 1,4-dibromobutane (DiBrb), petroleum ether (PE) and N-hexane (N-hex) (Fig. 2f and Supplementary Fig. S33). For example, no separation or displacement phenomenon has been observed in a long-term adhesion test for at least 14 days in N-hex (Supplementary Fig. S33). In addition, the adhesion strengths of adhesive remain relatively stable for 14 days after immersion in these organic solvents (Fig. 2f).

According to your suggestion, we have also investigated the resistance of SSFP adhesive to solvents like water, THF and DMF. Taking water as an example, minor amount of SSFP adhesive was dissolved (only 0.76%) after immersion in water for 5 days at room temperature by detecting the UV-vis absorption of SiW₁₂ (Supplementary

Figs. S36 and S37). Nevertheless, it became soft and partly transformed into traditional solvent-assisted adhesive, which would be attributed to the interaction of water molecules with the adhesive that results in parts of the hydrogen-bonded networks in SSFP adhesive to be collapsed and rebuilt. Furthermore, the adhesion strengths of SSFP adhesive at different relative humidity (from 40% to 80%) have also been tested, and the results show that the adhesion strengths of SSFP adhesive only have slight decrease with the increase of relative humidity (Supplementary Fig. S38), showing its high resistance to humidity.

Now we have added relative results and discussion in the revised manuscript (page 6, line 31-36, 41-47; page 7, line 1-4) and Supporting Information (page 39, 42-44).

Fig. 2. Morphology and strength property of SSFP adhesive. a) SEM and corresponding elemental mapping images of SSFP adhesive. b) Macroscopic photographs of SSFP adhesive obtained in kilogram scale and shear strength test. c) Photograph of the weights bonded by SSFP adhesive. d) Emergency leakage test performed using SSFP adhesive. e) Adhesion strengths of SSFP adhesive on various substrates. f) Adhesion strengths of SSFP adhesive on the interfacial adhesion system in various organic solvents for 14 days.

Supplementary Figure 33. Photographs of the SSFP adhesive after immersion in organic solvents (*N*-hex) for 14 days.

Supplementary Figure 36. Digital photographs of the adhesive after soaking in different solvents for 5 days. a) Water. b) THF. c) DMF.

Supplementary Figure 37. The SiW₁₂ leaching test of SSFP adhesive by soaking in water. a) UV-vis spectrum of SiW₁₂ in water. b) The plot of absorbance change at 263 nm upon the concentration increase of SiW₁₂. c) The corresponding dissolution ratio of SiW₁₂ in SSFP adhesive after soaking in water for different time.

Supplementary Figure 38. Adhesion strengths of SSFP adhesive at different relative humidity.

Reviewer 2:

A low-temperature tolerant adhesive was prepared through a heat-assisted process in this work. Because of hydrogen bonds between $\text{H}_4\text{SiW}_{12}\text{O}_{40}$ and polyethylene glycol, solvent-free polymer (SSFP) adhesive showed an adhesion strength of 3.7 MPa towards stainless steel and 2.96 MPa at $-196\text{ }^\circ\text{C}$. However, the adhesion strength of SSFP adhesive was not high enough for engineering application, and the advantages of using polyoxometalates and solvent-free process have not been fully demonstrated. In summary, this work does not meet the requirements of innovation and importance to be recommended for publication in Nature Communications. Some comments are listed below.

Response: Thanks for your kind comments. For the adhesion strength of SSFP adhesive, we have provided a detailed and in-depth discussion in the following response to comment, and find that the SSFP adhesive can also meet some special application scenarios. Besides, for the advantages of using POMs and solvent-free process, we have added the corresponding discussion, which are summarized into the following three points in detail, respectively. The advantages of using POMs: 1) favorable interface adhesion might be promoted by using POMs to regulate the crosslink density of polymer; 2) cohesion strength of adhesive would be markedly enhanced by the interaction with POMs; 3) POMs can be act as ideal templates for the theoretical calculations to elucidate the cohesion interaction mechanism at ultra-low temperature. The advantages of solvent-free processing process: 1) easy storage, transportation and processes in practical applications; 2) no emission of additional solvents and toxic exhaust gases after processing; 3) favorable for ultra-low temperature resistance and cyclic utilization. Therefore, combining the advantages of using POMs and solvent-free processing process, an ultra-low temperature resistant adhesive can be achieved and exhibit good adhesion strength. This is the first case of solvent-free processed POMs based adhesive with good adhesion strength, organic solvent stability, wide tolerable temperature range (i.e. -196 to $55\text{ }^\circ\text{C}$), long-lasting adhesion effects (> 60 days) at $-196\text{ }^\circ\text{C}$ and a specific cohesion interaction mechanism well-illustrated with the

assistance of DFT calculations. Besides, the raw materials are readily available and relatively cheap, and manufacturing at scale can be easily achieved by the facile scale-up solvent-free processing. To conclude, the slight change in preparation method by a solvent-free processing makes big differences in low temperature resistant performance, enabling us to pass the roadblock of ultra-low temperature resistance for the achieved adhesive. In the revised version, we have also added sufficient experiments (e.g., the influence of different polymers and molecular weight of PEG on the adhesion strength; the solvent stability of SSFP adhesive; the stability of SSFP adhesive at different relative humidity; the anti-freezing test of SSFP adhesive based on SiW₁₂ and PEG₂₀₀₀₀), characterizations (e.g., shear stretch test; UV-vis analysis) and theoretical calculations (e.g., the interaction energy between POMs and PEG by DFT calculations; the interfacial adhesive energy between SSFP adhesive and SS substrate at 25, 55 and -196 °C by MD simulation; the comparison of interaction energy of SSFP adhesive and SSFP/substrate with different number of simulated molecules), and provided intensive discussions to largely improve the quality of this work. We believe the quality of this work is largely enhanced during this process that can meet the high criteria of this journal and hope you agree.

1. SSFP adhesive showed an adhesion strength of 3.7 MPa towards stainless steel and 2.96 MPa at -196 °C. However, the adhesion strength of SSFP adhesive was much lower than that of structural adhesive (generally higher than 15 MPa), which is not enough for application in Arctic/Antarctic or planetary exploration.

Response: Thanks for your valuable comment. The adhesion strength of SSFP adhesive is indeed not as strong as that of structural adhesives. There are some engineering applications such as the structure bonding of spacecraft that do require structural adhesives with high adhesion strength. Nevertheless, most of the structural adhesives are prepared by solvent-assisted methods, which would result in many inconveniences in practical applications including storage, transportation, or processing processes, as well as the possible contaminating caused by the residue of solvents.

In this work, the SSFP adhesive with an adhesion strength of 3.7 MPa and 2.96 MPa at -196 °C towards stainless steel is sufficient to satisfy some practical applications. Actually, some adhesives with moderate adhesion strength are also applied in some special application scenarios, such as space electronics, space robots, spacesuit and inner decoration of spacecraft, etc. In addition, we know that the common temperature range at the North or South Poles is from -49 to -31.4 °C, and the maximum temperatures at the North and South Poles are 38 °C and 17.5 °C, respectively, which is among the applicable range of SSFP adhesive. Notably, the adhesion strength of SSFP adhesive is comparable or even superior to currently available commercial adhesives, such as solvent-free EVA (~3.8 MPa), 3M instant adhesive (~2.3 MPa) and double-sided tape (~0.25 MPa) (*Sci. Adv.* **4**, eaat8192, (2018); *J. Adhesion Sci. Technol.* **17**, 1831-1845 (2003)). Meanwhile, compared with solvent-free EVA, the SSFP adhesive exhibits better ultra-low temperature resistance. Besides, the adhesion strength of SSFP adhesive is superior to almost all of POMs based adhesives that have been reported so far (Supplementary Table 1). Furthermore, the SSFP adhesive has above-average

adhesion strength in the representative solvent-free adhesives reported at present (Supplementary Table 2). Therefore, the prepared SSFP adhesive can meet some working conditions in Arctic/Antarctic or planetary exploration. Even so, the SSFP adhesive still has some drawbacks that make it difficult to satisfy all the applications in Arctic/Antarctic or planetary exploration. To cover more application scenarios, we would develop higher performance adhesive through synthesizing and screening polymers and POMs types, as well as the processing technology of adhesives in our future work.

Now we have added the corresponding comparison of representative solvent-free adhesives and discussion in the revised manuscript (page 4, line 28, 29; page 5, line 1) and Supporting Information (page 66-67).

Supplementary Table 1. Performance comparison of the reported POMs based adhesives, and the test method is the same with all the references (lap joint).

Sample	POM	Cross-linked compound	Type	Temperature (°C)	Substrate	Adhesion strength (kPa)	Ref.
SSFP	SiW ₁₂	PEG (10 kDa)	Solvent-free	-196 RT	SS	2741.0 3736.0	This work
POMs/Pep	BW ₁₂ / PMo ₁₁ V	Ac-KKNSQCC- NH ₂ /GHK	Solvent-assisted (H ₂ O)	RT	Ti	82.5	12
SiW-PEG	SiW ₁₂	PEG (20 kDa)	Solvent-assisted (H ₂ O)	RT	SS	74.2	13
Pep1/SiW ₁₁ /H ⁺ Pep1/SiW ₁₁ /M ⁿ⁺	SiW ₁₁	GHK/H ⁺ GHK/Co ²⁺	Solvent-assisted (H ₂ O)	RT	Ti	36.5 21.1	14
P-PA55-PW30	PW ₁₂	Polyacrylate (100 kDa)/H ₃ PO ₄	Solvent-assisted (CH ₃ COCH ₃)	RT	Glass	211.0	15
PEI-PW ₁₂	PW ₁₂	PEI (1.8 kDa)	Solvent-assisted (H ₂ O)	RT	Wood	319.0	16
Ca-POM SNWs	PW ₁₂	Oleylamine/Ca ²⁺	Solvent-assisted (H ₂ O)	-196 100	SS	2160.0 ~ 1700.0	17
Pep1/SiW	SiW ₁₂	Ac- EEMQRAD- NH ₂	Solvent-assisted (H ₂ O)	RT	PEEK	29.6	18
GSSG/HPW	PW ₁₂	GSSH	Solvent-assisted (H ₂ O)	RT	Ti	53.3	19
NA/HP ₂ W ₁₈	P ₂ W ₁₈	3-(2-naphthyl)-l- alanine	Solvent-assisted (H ₂ O)	RT	PEEK	14.7	20
His/SiW	SiW ₁₂	Histidine	Solvent-assisted (H ₂ O)	RT	Wood	436.7	21
CS/SiW-PAM	SiW ₁₂	Polyacrylamide/ Chitosan (3 kDa)	Solvent-assisted (H ₂ O)	RT	PP	7.0	22

The corresponding references are provided at Supporting Information.

Supplementary Table 2. Comparison of representative solvent-free adhesives in terms of their adhesion strength, low temperature tolerance and temperature tolerance range.

Sample	Adhesion strengths (MPa)	Low temperature tolerance (MPa)	Temperature tolerance range (°C)	Ref.
SSFP	3.70	2.96 (-196 °C)	-196-55	This work
SEA0.2	10.2	N/A	N/A	23
DESPs	6.57	1.30 (-80 °C)	-80-80	24
Poly(TtADO-TA)-2	4.55	N/A	N/A	25
CT-2	4.40	>1.0 (-80 °C)	N/A	26
Poly(TA-DIB-Fe) 50:1	2.50	N/A	0-60	27
PC10-W1	2.49	1.17 (-196 °C)	-196-25	28
Ca-POM SNWs	2.16	1.70 (-196 °C)	-196-100	17
6-HTPB	2.14	0.50 (-18 °C)	-80-80	29
Pt-B21C7-II	1.90	N/A	N/A	30
DPETI	1.84	2.22 (-196 °C)	N/A	31
AZO-P1	1.34	N/A	25-50	32
CA/PEG ₂₀₀₀	0.53	0.85 (-196 °C)	-196-25	33
TADP30	0.36	N/A	25-50	34
P-PA55-PW30	0.21	N/A	N/A	35
LTFe	0.02	N/A	N/A	36

The corresponding references are provided at Supporting Information.

2. SSFP adhesive could not tolerate temperatures above 55 °C, which seriously affected the stability of SSFP adhesive during application.

Response: Thanks for your valuable comment. The tolerate temperature of adhesive is indeed an important parameter for practical applications. In this work, the maximum tolerated temperature of SSFP adhesive is 55 °C due to the limitation of the melting point of PEG ($T_g < 60$ °C, Supplementary Fig. S40). Actually, we have also tried to expand the high temperature resistance of SSFP adhesive. After tremendous experiments, we find that the maximum tolerated temperature of SSFP adhesive can exceed 55 °C or even to 110 °C by selecting higher molecule weight PEG ($M_n \geq 20000$) or adjusting the mass ratio of POMs to PEG (POMs : PEG $\geq 3:1$), yet these conditions will also obviously weaken the viscosity of SSFP adhesive, which is not conducive to the surface spreading or gap-filling during the adhesive applications. The SSFP adhesive has a wide temperature resistance range (-196 to 55 °C), which exceeds most of POMs based adhesives (Supplementary Table 1) and solvent-free adhesives (Supplementary Table 2). Remarkably, the SSFP adhesive in this work can be used in liquid nitrogen (-196 °C), which can be regard as the best POMs based solvent-free adhesives reported to date. In follow-up work, we need to finely tune the type,

molecular weight of polymers, as well as the ratio of building blocks to simultaneously improve the adhesion strength, viscosity and temperature resistance range of adhesive, and related works will be reported sooner or later.

Now we have added related discussion and the comparison tables in the revised manuscript (page 4, line 28, 29; page 5, line 1) and Supporting Information (page 46, 66 and 67).

3. Water is more common than organic solvents in daily life and in Arctic/Antarctic. The water resistance of SSFP adhesive should be investigated.

Response: Thank you for your insightful suggestion. According to your suggestion, we have investigated the resistance of SSFP adhesive to water through both water immersion and humidity resistance experiments. By detecting the UV-vis absorption of SiW₁₂ in the soaked solution, we found that only minor amount (0.76%) SSFP adhesive was dissolved into the water after soaking for 5 days (Supplementary Figs. S36 and S37). Even though, it became soft and was partly transformed into traditional solvent-assisted adhesive, which would be attributed to the interaction of water molecules with the adhesive that results in parts of the hydrogen-bonded networks in SSFP adhesive to be collapsed or rebuilt. Furthermore, the adhesion strengths of SSFP adhesive at different relative humidity (from 40% to 80%) have also been tested, and the results show that the adhesion strengths of SSFP adhesive only have slight decrease with the increase of relative humidity (Supplementary Fig. S38), showing its high resistance to humidity.

Now we have added relative results and discussion in the revised manuscript (page 6, line 41-47; page 7, line 1-4) and Supporting Information (page 42-44).

Supplementary Figure 36. Digital photographs of the adhesive after soaking in water for 5 days.

Supplementary Figure 37. The SiW₁₂ leaching test of SSFP adhesive by soaking in water. a) UV-vis spectrum of SiW₁₂ in water. b) The plot of absorbance change at 263 nm upon the concentration increase of SiW₁₂. c) The corresponding dissolution ratio of SiW₁₂ in SSFP adhesive after soaking in water for different time.

Supplementary Figure 38. Adhesion strengths of SSFP adhesive at different relative humidity.

Reviewer 3:

Adhesives that can withstand cryogenic temperatures are in great demand for material bonding in Arctic/Antarctic and outer space exploration, but their supply is scarce. In this study, the authors prepared kilogram-scale, solvent-free polymer adhesives based on POM by a heat-assisted process. The resulting SSFPs exhibit excellent interfacial adhesion capabilities on a variety of substrates and are notable for their high bond strength and organic solvent stability, wide allowable temperature range (i.e., -196 to 55 °C), and long-term adhesion effectiveness (> 60 days) at -196 °C. The high performance of SSFPs is superior to commercial hot melt adhesives commercial hot melt adhesives. Furthermore, experimental results combined with theoretical calculations indicate that the strong interaction energy between POM and PEG is the main reason for the high adhesive performance at low temperatures with increased cohesive strength, reduced polymer crystallization, and volume shrinkage. This research sheds light on the development of more types of low temperature resistant adhesives and advanced solvent-free adhesives for Arctic and Antarctic and planetary exploration. The topics covered in this manuscript range from basic to applied, and are sure to be of interest to a diverse readership. Therefore, this manuscript should be accepted by Nature Communications. I would, however, like to express some concerns as follows. I hope the authors will address them appropriately before publication.

1. The phrase "H protons" appears several times, but I think just "protons" is fine.

Response: Thank you for your kind suggestion. We have modified the inappropriate descriptions of "H protons" to "protons" and checked through all of the contents to avoid similar problems.

Now we have added related modification in the revised manuscript (page 2, line 28; page 6, line 8, 11 and 26; page 10, line 22).

2. The relationship between OPLS and UFF used in the MD simulation is unclear. Please explain in more detail.

Response: Thank you for your kind suggestion. There are both metallic and non-metallic elements in SSFP adhesive, so it is necessary to use the OPLS force field with the assistance of UFF force field when simulating such a complex system. The detailed description of OPLS and UFF used in the MD simulation is updated in this revised version. The OPLS force field is developed by Jorgensen et al. at Yale University and is specially designed to emphasize condensed phase simulations (*J. Am. Chem. Soc.* **1988**, *110*, 1657-1666.; *J. Am. Chem. Soc.* **1996**, *118*, 11225-11236.). This force field is suitable for condensed phase simulation polymers such as proteins, sugars, and organic small molecules, yet it does not support metal elements. In addition, UFF is a universal force field covering the entire periodic table (*J. Am. Chem. Soc.* **1992**, *114*, 10024-10035.). It has the same force field function form as OPLS, yet the applicable scope is different and more suitable for metallic elements such as Ni, Cr, W and Mn. Therefore, OPLS force field is assisted with UFF force field to simulate the complex system of SSFP adhesive.

Now we have added related description in the revised Supporting Information (page

6).

3. Is it safe to use only NVT ensemble simulations? NVT would not be able to reproduce the change in density with temperature change. It would be better to do NPT ensemble as well.

Response: Thanks for your constructive suggestion. We referred to relevant references and found that NVT was widely used to analyze similar situations (e.g., refs. *J. Am. Chem. Soc.* **2020**, *142*, 21522-21529.; *J. Am. Chem. Soc.* **2022**, *144*, 16389-16394.; *CCS Chem.* **2020**, *2*, 1690-1700.). We also strongly agree with the reviewer's suggestion that NPT can better simulate the equilibrium conditions. Therefore, we additionally calculated the simulation of NPT ensemble (Supplementary Fig. S58 and movies S7-to-S9). In the comparison of the equilibrium states of production with different temperatures (i.e. 25, 55, and -196 °C), the results of NPT simulation are basically consistent with previous calculations by NVT ensemble.

Now we have added the related discussion and results in the revised manuscript (page 10, line 23-25) and Supporting Information (page 6, 64 and 65).

Supplementary Figure 58. Adhesive mechanism and MD simulation based on the NPT ensemble. a) MD simulations of configurations of molecular models of SSFP and SS substrate. b) The interaction energy and the ratio of formative hydrogen bonds between

SSFP adhesive and SS substrate at 25, 55 and -196 °C. c) The comparison of interaction energy and the ratio of formative hydrogen bonds between SSFP and PSFP adhesive on SS substrate.

4. I have understood the cohesive strength of the interaction between POM and PEG. However, experimentally, the authors have observed fracture at the adhesive interface, so it would be good to evaluate not only the cohesive energy but also the adhesive energy between the substrate and the adhesive in the simulation and compare it with the cohesive interaction energy.

Response: Thanks for your kind suggestion. We have employed NPT ensemble to analyze the interaction energy between the three components of POM, PEG and the substrate. Stainless steel is chosen as the model substrate (referring to the research by Wang et al., *J. Am. Chem. Soc.* **2022**, *144*, 16389-16394.). The stable configurations of molecular model of SSFP adhesive and SS substrate are shown in Supplementary Figs. S58 and movies S7 to S9. Meanwhile, the MD simulations verify that there is favorable interfacial adhesion between SSFP adhesive and SS substrate, and the interaction energy increases gradually as the temperature decreases. which will effectively inhibit volumetric contraction and enhance mechanical force transmission across the substrate, endowing the SSFP adhesive with good adhesion performance (Supplementary Figs. S58 and movies S7 to S9).

For the comparison between cohesive energy and interfacial adhesive energy, it is relatively difficult to accurately compare them through the simple theoretical simulation, because the number of POMs, PEG and substrate molecules in the simulation process will directly affect the magnitudes of cohesive energy and interfacial adhesive energy. Additionally, the surface roughness of substrate, the thickness and area of adhering coating, as well as surface spreading of adhesive are not easily simulated accurately. All of these will obviously affect the simulation results of interaction energy. Here, we evaluated cohesive energy and adhesive energy between substrate and adhesive (30 of POMs and 150 of PEG in our simulations). The results showed that the cohesive energy was low than the interfacial adhesion energy (Figure R2). Interestingly, the cohesive energy of adhesive increases with the number of simulated molecules (POMs and PEG), yet the interfacial adhesive energy is almost constant at the same stainless steel substrate (Figure R3). To get closer to the real adhesion situation, more complicated or bigger simulation system might need to be built in the structures of MD simulation to demonstrate the accurate results of interfacial adhesion energy, which still remains a giant challenge for the theoretical simulation at present. At present, most of reported references also failed to achieve this comparison (*J. Am. Chem. Soc.* **2020**, *142*, 21522-21529.; *J. Am. Chem. Soc.* **2022**, *144*, 16389-16394.; *CCS Chem.* **2020**, *2*, 1690-1700.) through the theoretical simulation and this will be the target of our future work.

Now we have added the related discussion and results in the revised manuscript (page 10, line 23-25) and Supporting Information (page 6, 64 and 65).

Supplementary Figure 58. Adhesive mechanism and MD simulation based on the NPT ensemble. a) MD simulations of configurations of molecular models of SSFP and SS substrate. b) The interaction energy and the ratio of formative hydrogen bonds between SSFP adhesive and SS substrate at 25, 55 and -196 °C. c) The comparison of interaction energy and the ratio of formative hydrogen bonds between SSFP and PSFP adhesive on SS substrate.

Figure R2. The comparison of cohesive energy and adhesive energy between substrate and adhesive. a) SSFP adhesive and SSFP/substrate. b) PSFP adhesive and PSFP/substrate.

Figure R3. The comparison of cohesive energy and adhesive energy between substrate and adhesive. a) The interaction energy of SSFP adhesive and SSFP/substrate with different number of simulated molecules. b) The variation curves of average interaction energy of SSFP adhesive and SSFP/substrate with the number of simulated molecules. (the structures of MD simulation are consist of 30 H₄SiW₁₂O₄₀ and 150 PEG fragments. (50 H₄SiW₁₂O₄₀ and 250 PEG fragments; 100 H₄SiW₁₂O₄₀ and 500 PEG fragments) Meanwhile, the simulated molecules of SS substrate remain the same).

5. It is clear from the MD simulation movies that the adsorption conformation of PEG around POM is really diverse; did the authors take that diversity into account in their DFT calculations?

Response: Thank you for your kind suggestion. In this new version, we take the diverse adsorption conformation of PEG around POM into account in their DFT calculations. In detail, we investigate the interaction between POMs and PEGs by considering the motion of the molecules in an equilibrium environment with time. The radial distribution function (RDF) of the center of mass (com) of POMs and the com of PEGs are statistically calculated for the production of 1 ns. The RDF presents a broad peak, indicating that there is no clear PEG layer around the POM. The coordination number (CN) between POMs and PEGs is obtained by integration. When CN = 1, the average distance between the com of POM and PEG is 8.02 Å, which is consistent with the structures of POM and PEG and the distance of interaction between them. In addition, when CN = 2 or 3, the radius of effect is respectively 8.52 Å and 8.88 Å. We perform DFT using RDF as a reference for selecting the models of POM and PEG in 1 : 1, 2 : 1 and 3 : 1 to further investigate their interactions. A total of thirteen structures are investigated through DFT theoretical calculations, including six structures with POM : PEG = 1 : 1, four structures with POM : PEG = 1 : 2, and three structures with POM : PEG = 1 : 3. The results reveal that the interaction energy (Fig. 4a) between them is -287.7 kJ/mol for one PEG fragment, -537.4 kJ/mol for two PEG fragments, and -826.7 kJ/mol for three PEG fragments, respectively.

Now we have replaced the DFT calculation results and added relative discussion in the revised manuscript (page 9, line 13; page10, line 1-3) and Supporting Information (page 55-56).

Supplementary Figure 49. The radial distribution function (RDF) of POMs and PEGs for the production of 1 ns.

Supplementary Figure 50. Independent gradient model based on Hirshfeld partition (IGMH) and interaction energies (ΔE , kJ/mol) between SiW₁₂ and PEG for the adhesive formation. a) SiW₁₂ and one PEGs. b) SiW₁₂ and two PEGs. c) SiW₁₂ and three PEGs.

REVIEWERS' COMMENTS

Reviewer #1 (Remarks to the Author):

The authors have responded well to the previous critiques. This manuscript could be published in this journal.

Reviewer #2 (Remarks to the Author):

The revised manuscript can be accepted.

Reviewer #3 (Remarks to the Author):

The authors performed additional calculations, thoroughly researched the relevant literature, and sincerely addressed my concerns. Therefore, I find the authors' revision work to be adequate. I recommend that the manuscript be accepted as is.